# Influence of Temperature on Mechanical Properties of Nanocrystalline 316L Stainless Steel Investigated via Molecular Dynamics Simulations

**DOI:** 10.3390/ma13122803

**Published:** 2020-06-22

**Authors:** Abdelrahim Husain, Peiqing La, Yue Hongzheng, Sheng Jie

**Affiliations:** 1State Key Laboratory of Advanced Processing and Recycling of Nonferrous Metals, Lanzhou University of Technology, Lanzhou 730050, China; abdosh.husian@yahoo.com (A.H.); zhengyuehong1986@126.com (Y.H.); shengj605@163.com (S.J.); 2Department of Physics, Faculty of Science and Technology, University of Shendi, Shendi, River Nile State, P.O. Box 407, Sudan

**Keywords:** mechanical properties, 316L austenitic stainless steels, grain size, temperature effect, molecular dynamics, embedded atom method

## Abstract

Molecular dynamics simulations were conducted to study the mechanical properties of nanocrystalline 316L stainless steel under tensile load. The results revealed that the Young’s modulus increased with increasing grain size below the critical average grain size. Two grain size regions were identified in the plot of yield stress. In the first region, corresponding to grain sizes above 7.7 nm, the yield stress decreased with increasing grain size and the dominant deformation mechanisms were deformation twinning and extended dislocation. In the second region, corresponding to grain sizes below 7.7 nm, the yield stress decreased rapidly with decreasing grain size and the dominant deformation mechanisms were grain boundary sliding and also grain rotation. The yield strength and Young’s modulus were both found to decrease with increasing temperature, which increased the interatomic distance and thereby decreased the interatomic bonding force.

## 1. Introduction

Austenitic stainless steels are utilized in numerous fields due to their outstanding corrosion and oxidation resistance and excellent formability [1]. A great deal of experimental research has been conducted in an effort to improve the mechanical properties of stainless steels. For example, Pakiela et al. [2] prepared a nanostructured 316L stainless steel with a grain size of 65 nm via high-pressure torsion and reported a tensile strength of approximately 1340 MPa. However, both twinning and martensite transformation occurred during the grain refinement process. Chen et al. [3] studied the room-temperature tensile properties of a nanocrystalline 316L austenitic stainless steel with a mean grain size of 40 nm equipped via surface mechanical attrition treatment. It was reported that this sample showed an extremely high yield strength of 1450 MPa and followed the Hall–Petch relationship extrapolated from the coarse grains.

Many plastic deformation ways [4], for example, equal channel angular processing, accumulative roll bonding, high-pressure torsion, mechanical milling, cold rolling, and ball milling, can be applied to lessen the grain size to the nanoscale. In such configurations, the nanoscale grain boundaries, which play very significant protagonists in determining the mechanical properties of nanocrystalline materials, are typically in non-equilibrium conditions or, more strictly, they are diffuse boundaries between dislocation cells. Various other defects (e.g., point defects, twinning, stacking faults, and dislocations) are also regularly seen in structures produced via severe plastic deformation [4]. Our group has performed several studies aimed at improving the mechanical properties of 316L stainless steel and exploring the role of microscale or nanoscale structures [5,6,7,8].

Numerous phenomena occur in the microstructure of nanocrystalline stainless steels, such as stacking faults, dislocations, and twinning, and these types of deformation impact the mechanical properties of the nanocrystalline compounds. However, experimental observation of the mechanisms underlying the development of twinning and dislocations during the deformation process is difficult. Molecular dynamics simulations are an excellent choice for clarifying the microstructural changes that occur during the plastic deformation of nanocrystalline materials by a grain size of less than 10 nm. In previous studies of the mechanisms of deformation and microscopic evolution of nanocrystalline stainless steel, some simulations have indicated the ability of the grain boundaries in nanocrystalline samples to accommodate externally applied stress through grain boundary sliding [9] and the emission of partial dislocations [10]. There is no previous research of nanocrystalline 316 stainless steel, but by benefiting from some previous studies of the mechanical properties of nanocrystalline steels and ferrite alloys, the mechanical properties of stainless steel can be studied. Zheng et al. [11] used molecular dynamics simulations to investigate the roles of grain boundaries and dislocations in nanocrystalline copper at various stages of deformation. For small grain sizes, the dominant deformation mechanisms were found to be grain boundary sliding, grain boundary rotation, and the emission of partial dislocations (Shockley dislocations) from grain boundaries into grains to leave behind stacking faults. For large grain sizes, the dominant deformation mechanisms were extended dislocations and grain boundary rotation. The flow stress of nanocrystalline copper was reported by J. Schiøtz. [12] to be top at an average grain size of 10–15 nm. Below this range, the flow stress decreased with decreasing average grain size, and the opposite trend was observed when the average grain size was above a certain critical value. Li et al. [13] also employed molecular dynamics simulations to explore plastic deformation in nanocrystalline aluminum samples with average grain sizes of 10, 20, and 30 nm. Competition between the grain-boundary-mediated and dislocation-mediated mechanisms was observed during the recovery of the plastic strain. Ding et al. [14] used molecular dynamics to examine the influence of the grain size on the microscopic deformation mechanism of polycrystalline TiAl alloys. For small grain sizes, the yield stress of the nanocrystalline TiAl alloys was found to increase with increasing grain size, and the dominant plastic deformation mechanisms were grain boundary migration besides grain rotation. Above a certain critical grain size, the yield stress decreased with increasing grain size and the dominant plastic deformation mechanisms were dislocation slipping besides deformation twinning. Li et al. [15] explored the mechanical properties of nanocrystalline platinum via molecular simulations and determined that the critical mean grain size was approximately 14.1 nm. Above this critical grain size, the dominant deformation mechanism was dislocation motion, whereas below the critical grain size, it was grain boundary sliding. The Young’s modulus, and all mechanical properties, decreased with increasing temperature.

On the basis of previous molecular dynamics simulations [16,17], plastic deformation is controlled by grain-boundary-mediated processes for very small grain sizes and by dislocation-mediated processes for huge grain sizes.

In this work, MD simulations were performed to investigate the mechanical properties of nanocrystalline 316L stainless steel, including the flow stress, yield stress, ultimate strength, and Young’s modulus, and their dependence on the medium grain size from 2.5 to 11.5 nm. Furthermore, the effects of the grain shape and temperature on the mechanical properties of crystalline stainless steel were also considered. The obtained results are expected to prove valuable for the development of nanocrystalline 316L stainless steels exhibiting ultrahigh strength and good ductility.

## 2. Simulation Methodology

The large-scale atomic/molecular massively parallel simulator (LAMMPS) molecular dynamics program [18] and the Atomsk software [19] were utilized to construct four cubic samples with dimensions of 200 × 200 × 200 Å and average grain sizes of 11.5, 9.9, 7.7, 4.1, 3.8, or 2.5 nm. The grain boundaries were generated using the Voronoi construction method [20], as depicted in Figure 1. As shown in Figure 1a, the model was composed of atoms of three elements, namely, nickel (12 wt. %), chromium (17 wt. %), and iron (71 wt. %), which is similar to the composition of 316 stainless steel. In Figure 1c, the face-centered cubic (fcc) lattice structures of the grains are indicated in green and the close-packed (cp) grain boundaries are indicated in white. The embedded atom method potential developed by Zhou et al. [21] was adopted to analyze the atomic interactions in the fcc nanocrystalline austenitic stainless steel. After constructing the simulation boxes and prior to lattice, periodic boundary conditions were applied in the *x*, *y*, and *z* directions and the models were minimized using the conjugate gradient algorithm to obtain a stable atomic configuration. Finally, the systems were relaxed under a pressure of 0 bar and temperature of 300 K using the Nose–Hoover isobaric–isothermal ensemble (NPT) [22] for 200 ps with a time step of 2 fs. After relaxation, the bulk nanocrystalline Fe-Cr-Ni austenitic stainless steel was subjected to uniaxial tensile deformation at room temperature with a constant strain rate of 1.0 × 10^10^ s^−1^ in the *x* direction.

The atomic structure during the simulation was visualized using the OVITO software [23]. The common neighbor analysis (CNA) technique was used to visualize the crystal defects and local atomic structure [24]. As shown in Figure 1b, the various types of crystal defects and atoms were indicated in different colors, such as green atoms for fcc structures, blue atoms for body-centered cubic (bcc) structures, red atoms for stacking faults, and white atoms for grain boundaries or dislocation cores.

## 3. Results and Discussion

### 3.1. Influence of Grain Size on Mechanical Properties

Figure 2 presents the simulated stress and strain curves for the nanocrystalline austenitic stainless steel samples with different grain sizes. All simulations were performed at a temperature of 300 K under a strain rate of 1 × 10^10^ s^−1^. It can be seen that the elastic modulus decreased with declining grain size when the average grain size was lower than 7.7 nm. Table 1 summarizes the mechanical properties of the nanocrystalline austenitic stainless steel samples, including the ultimate strength, yield stress, and Young’s modulus. It can be seen that the strength decreased with increasing grain size for average grain sizes above 7.7 nm and increased with increasing grain size intended for average grain sizes below 7.7 nm.

To assess whether the samples exhibited a Hall–Petch relationship, the yield stress at 300 K was plotted versus the reciprocal grain size (*d*^−1/2^). As shown in Figure 3, the data could be fitted by two straight lines. For the red line, the yield stress increased with growing grain size, indicating an inverse Hall–Petch relationship below an average grain size of 7.7 nm. For the black line, the yield stress decreased with growing grain size, indicating a conventional Hall–Petch relationship above an average grain size of 7.7 nm.

The influence of the grain size on the yield strength of the nanocrystalline stainless steel samples was examined in more detail. Figure 4 shows the twins and dislocations in samples possessing dissimilar grain sizes during plastic deformation. The grain boundary structure is colored gray, and stacking faults and twins are colored red. Figure 4a depicts the sample with a grain size of 2.5 nm. As the strain was increased, the stacking faults made by partial dislocation movement started to pile up. In contrast, pointing at a grain size of 7.7 nm (Figure 4b), new stacking faults emerged in the grains and the original stacking faults disappeared inside the grains. This phenomenon can be explained by the existence of a correlation between the grain size and the generation and disappearance of stacking faults. At grain sizes above 7.7 nm, the space needed for dislocation movement was very large, leading to a great possibility of dislocations being absorbed. This hindered the dislocation movement and increased the strain–stress curves. In contrast, at grain sizes below 7.7 nm, the partial dislocations could easily move to the grain boundary. Consequently, partial dislocations emitted from the grain boundary may possibly be hindered by the grain boundary. This, in turn, ran to a decrease in the strain–stress curve and therefore the strength.

To further analyze the plastic deformation behavior of the nanocrystalline stainless steel samples, we selected the samples with a mean grain sizes of 3.6 and 9.9 nm. For the sample with a mean grain size of 3.6 nm, which is below the critical grain size, major deformation occurred in the grain boundaries (sliding and rotation). When the strain was increased from 7% to 12.5%, the additional strain built up mainly in the grain boundaries (see Figure 5c). Dislocation motion was also observed, although it was not the dominant deformation mechanism.

For the sample with a mean grain size of 9.9 nm, a small amount of grain boundary sliding was observed, although the majority of the deformation occurred in the grain cores. Figure 5 shows the evolution of several representative dislocations, as indicated by the blue arrows, and a comparable phenomenon is common during deformation. All atoms in grain boundaries possess higher energies than those in grain cores. Under an applied load, the energy of grain boundaries gradually increases. To dissipate this excess energy, partial dislocations (Shockley dislocations) are emitted from the grain boundaries, as observed in Figure 5a,b. Some other types of dislocations, such as perfect, stair-rod, Frank, and Hirth dislocations, that remained within the grain boundaries and did not emit inside the grain, can also be observed in Figure 5c–f. As the strain was increased, the partial dislocations glided and moved through the grains, as shown in Figure 5c,d. Finally, the partial dislocations faded into the grain boundaries, leaving behind stacking faults within the grains as shown in Figure 5d. Furthermore, it can be observed that twins were formed directly from the grain boundaries via the emission of two types of parallel partial dislocations or the collision of two stacking faults, as displayed in Figure 5e.

For the sample with an average grain size of 3.6 nm, main deformation occurred in the grain boundaries. The dominant deformation mechanism for the small grains was grain boundary movement, such as grain rotation and grain boundary sliding. Figure 6 shows snapshots of the atomic configuration of this sample under various tensile loads (0, 0.05, 0.075, 0.1, 0.125, and 0.15), revealing the presence of both lesser grains and moderately large grains that underwent both growth and shrinkage with increasing strain. For example, the grain boundary between G1 and G4 was found to expand, which indicates the occurrence of grain boundary movement. With increasing strain, G1 grew in size while G3 shrank. As the strain was increased to 0.125, G3 gradually disappeared, as shown in Figure 6f, and the grain boundaries between G1 and G4 merged. Owing to this phenomenon of grain growth and shrinkage, the small grains underwent continuous shrinkage and merged with the larger surrounding grains. This phenomenon can be explained by the higher kinetic energies and more diffuse atoms of the grain boundaries between small and large grains. Therefore, the grain boundaries among small and large grains were more susceptible to migration. As the strain was increased, the atoms of the grain boundaries and grains moved in inconsistent directions and underwent mutual slipping, causing the grains to rotate under complex stress conditions. Besides, the phenomenon rotation and migration of the grain boundaries and the rotation inside of the atoms were seen.

In addition to grain movement, other defects appeared inside the grains, as seen in Figure 6e. As the strain was gradually increased, partial dislocations were emitted in G1. These partial dislocations grew from nuclei in the grain boundaries and then lengthy inside the grains. Lastly, these partial dislocations were shown via the grain boundary, leaving behind stacking faults in the grains. The number of stacking faults inside the grains increased with increasing sample strain, as did the number of local dislocations. At the same time, some stacking faults increasingly disappeared during the deformation way. Owing to the growth of some of the grains within the sample, we found that the involvement of partial dislocation to plastic deformation was minimal compared to grain boundary deformation. The reason for the transmission of grain boundaries was the movement of atoms on both sides of the grain boundary, which tended to be perpendicular to the grain boundary interface. Grain boundary migration, affected by the diffusion of atoms from one grain to another, led to either the growth or contraction of the grains.

The grain sliding and grain rotation were further analyzed. Figure 7 presents atomic displacement maps and the corresponding CNA images of the samples with an average grain size of 3.6 nm. In Figure 7a,c, the atomic motion from the initial configuration to a strain of 10% after eliminating homogeneous deformation is indicated by the arrows. The larger yellow arrows indicate the overall movement of the grain boundaries. Figure 7a,c show the deformation mechanisms of grain boundary sliding and grain rotation, respectively. As shown in Figure 8a, using the atomic displacement vectors, the plastic deformation mechanism was also found to occur in the sample with an average grain size of 2.5 nm as the strain was increased. Upon increasing the strain, grain boundary sliding, grain rotation, and grain migration were observed, as shown in Figure 8b.

### 3.2. Young’s Modulus

The elastic modulus of a material is an indicator of the elastic properties of the material under loading and hence, the atomic binding forces. The correlation between the Young’s modulus and the applied stress and strain is defined by Hooke’s law:(1)$$\sigma =E_{0}\varepsilon ,$$
where σ is the applied stress, ε is the strain, and $E_{0}$ is the Young’s modulus.

Figure 9 shows a plot of the Young’s modulus as a function of reciprocal grain size. It can be seen that the Young’s modulus increased with increasing grain size when the grain size was lower than 7.7 nm; a similar linear relationship was also suggested by Nan et al. [25]. Zhou et al. [26] and Rida et al. [27] used molecular dynamics simulations to explore the impact of grain size on the mechanical properties of nanocrystalline copper and found that the correlation between the modulus and grain size was generally linear. Sanders et al. used experimental measurements to estimate the Young’s modulus of nanocrystalline metals such as copper and palladium and reported that the Young’s modulus was proportional to the reciprocal grain size [28]. However, the correlation between *E* and grain size tends to be more suitable for nanocrystalline 316 austenitic stainless steel in the current study, at least in the range of average grain sizes between 3.6 and 7.7 nm.

### 3.3. Influence of Temperature on The Mechanical Properties

The mechanical properties of the nanocrystalline austenitic stainless steel samples were simulated at various temperatures (10, 100, 300, 600, and 900 K) to evaluate the influence of temperature. Figure 10 presents the stress–strain curves for the sample with an average grain size of 2.5 nm under different testing temperatures, and Figure 11 shows the yield stress as a function of temperature for this sample. It may be noted the yield stress increased with decreasing temperature. Increasing the temperature increases the interatomic distance, which leads to a lower yield strength. Figure 12 presents a plot of the Young’s moduli of the samples with average grain sizes of 2.5, 4.1, and 7.7 nm with respect to temperature. It may be noted the Young’s modulus decreased with increasing temperature for all three samples. The capability of atoms to diffuse increases with increasing temperature, and thus atoms are more easily displaced under application of an external load at higher temperatures. Consequently, a higher temperature increased the deformation resistance of the materials, thereby decreasing the Young’s modulus. The correlation between the Young’s modulus and temperature was approximately linear, at least in the temperature range of 10–900 K.

To further examine the influence of temperature on the plastic deformation of nanocrystalline stainless steel, we calculated the average grain boundary thickness for all samples at 300, 600, and 900 K. As shown in Figure 13, the grain boundary thickness increased with increasing temperature, owing to the movement of the disordered atoms to the grain boundary. Figure 14 presents snapshots of the polycrystalline stainless steel sample with an average grain size of 3.6 nm at 10, 300, and 900 K. The green- and white-colored atoms indicate grains and grain boundaries, respectively. As highlighted by the red, yellow, and black circles, the grain boundary thickness increased with increasing temperature. Furthermore, the influence of temperature on the grain boundary thickness was found to decrease with increasing grain size.

It is known that the plastic deformation of nanocrystalline stainless steel involves two mechanisms, namely, dislocation slip and twinning. Twinning and slipping both require the movement of dislocations. The plastic deformation observed in the current samples consisted of perfect, partial, and extended dislocations. Figure 15 presents snapshots of the sample with an average grain size of 7.7 nm at various temperatures, where the partial dislocations are indicated by arrows. As shown in Figure 15a, partial dislocations were first emitted from the grain boundary at a strain of 0.002. As the strain was increased, the partial dislocation vanished to leave behind a stacking fault, as indicated in Figure 15b. Upon further increasing the strain, the stacking faults accumulated and then moved, resulting in the twins indicated in Figure 15c. Comparison of the snapshots obtained under different temperatures revealed that increasing the temperature delayed the emission of partial dislocations from the grain boundaries under the same strain and for the same grain size.

The twins and stacking faults consisted of partial dislocations formed from the grain boundaries and are indicated by the red-colored atoms (hcp) in the snapshots. These were used to calculate the volume fraction of stacking faults and twins in the nanocrystalline stainless steel sample at various temperatures. As shown in Figure 16, the volume fraction of twins and stacking faults decreased with increasing temperature. Therefore, increasing the temperature increased the ability of the disordered atoms to move to the grain boundary and thereby decreased the deformation resistance of the sample, leading to a decreased concentration of stacking faults and twins and lower strength.

Figure 17 displays the correlation between temperature and dissociation density. The density of grains declined with increasing temperature. This value differed between high temperature (900 K) and low temperature (300 K). Because of the increased temperature, a large number of atoms can obtain enough energy to get over energy barriers, which affects rearrangement of atomic structures. The magnification of dislocation disappearing, and a decline in dislocation density. This eventually reduces the phenomenon that dislocation stacking increases the strength of materials. Wherefore, Figure 11 displays that as the temperature increases, the average yield stress of the 316 stainless steel alloys is reduced.

## 4. Conclusions

In the present study, the mechanical properties of nanocrystalline 316L stainless steel with various average grain sizes were simulated using the Atomsk software with the Voronoi construction method. Molecular dynamics simulations were applied to explore the mechanical properties and plastic deformation mechanisms of the samples under tensile deformation. The results of this study can be summarized as follows:(1)The results indicate a reflection of the conventional Hall–Petch relationship down to critical grain size of 7.7 nm with a critical average grain size of 7.7 nm and a temperature of 300 K.(2)For grain sizes below 7.7 nm, the yield stress gradually decreased with decreasing grain size. The dominant plastic deformation mechanisms were grain boundary sliding and grain rotation.(3)For grain sizes larger than 7.7 nm, the yield stress gradually increased with decreasing grain size. The dominant plastic deformation mechanisms were dislocation formation and deformation twinning.(4)As the temperature was increased, the mechanical properties (Young’s modulus, ultimate strength, yield stress, and flow stress) deteriorated.

The results of this study provide a deeper insight into the role of microstructure on plastic deformation and are expected to prove valuable in the design of stainless steels with desirable mechanical properties and nanocrystalline phases.

## Figures and Tables

**Figure 1 materials-13-02803-f001:**
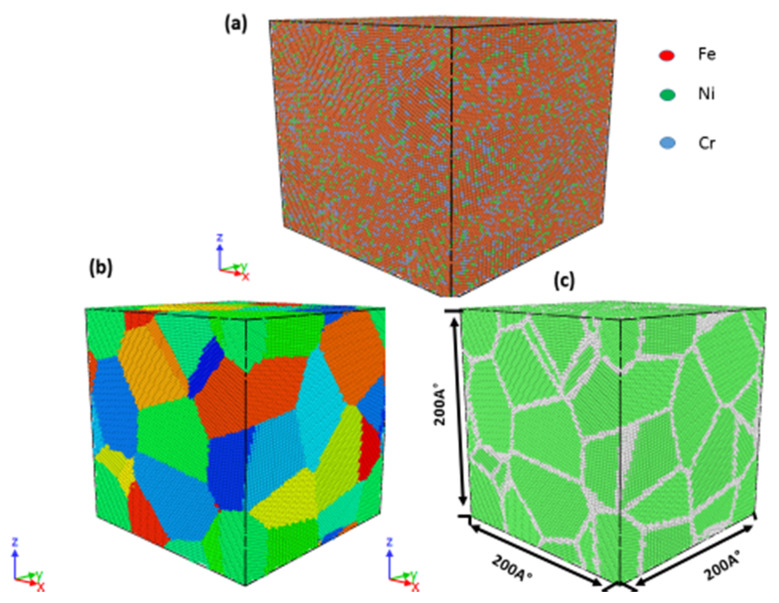
3D atomic configuration of nanocrystalline 316L stainless steel with an average grain size of 2.5 nm. (**a**) Atomic composition, (**b**) grain identity number, and (**c**) common neighbor analysis. Green atoms indicate fcc structures, blue atoms indicate bcc structures, red atoms indicate hexagonal close-packed (hcp) structures, and white atoms indicate other lattice structures.

**Figure 2 materials-13-02803-f002:**
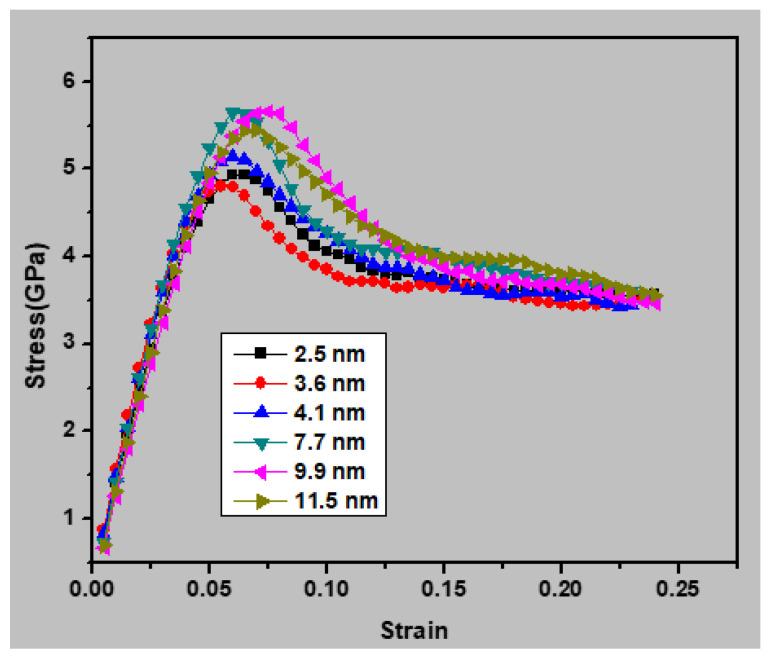
Simulated stress–strain curves for the nanocrystalline austenitic stainless steel samples with different grain sizes.

**Figure 3 materials-13-02803-f003:**
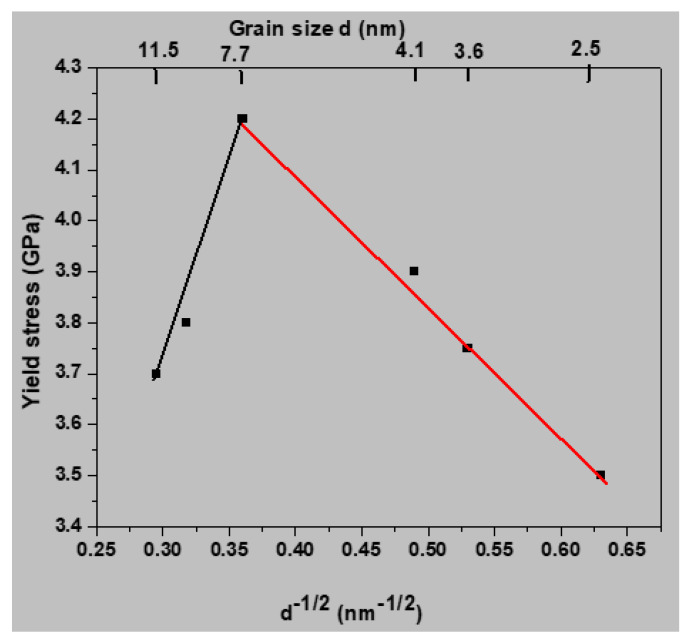
Influence of grain size on the yield stress at 300 K. The data were fitted by two straight lines.

**Figure 4 materials-13-02803-f004:**
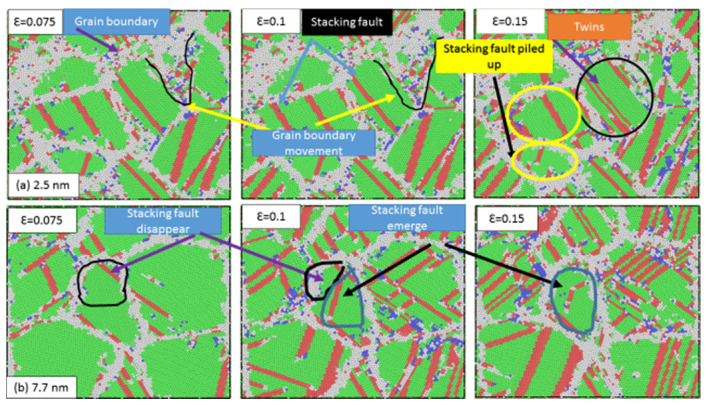
Twins and dislocations (accumulation and disappearance) in the samples with grain sizes of (**a**) 2.5 nm and (**b**) 7.7 nm during various stages of plastic deformation (ε = 0.075, ε = 0.1, and ε = 0.15).

**Figure 5 materials-13-02803-f005:**
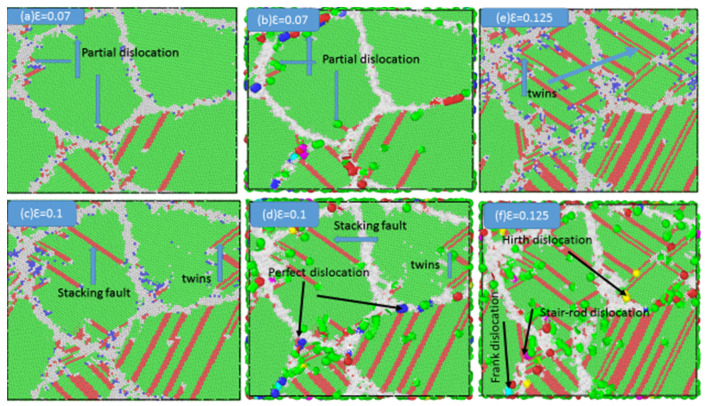
(**a**,**b**) Emission of partial dislocations from grain boundaries, (**c**,**d**) stacking faults in the grains, (**e**) twins formed by the emission of partial dislocations from grain boundaries, and (**f**) other types of dislocations for the sample with an average grain size of 9.9 nm under various strains: (**a**) ε = 0.07, (**b**) ε = 0.07, (**c**) ε = 0.1, (**d**) ε = 0.1, (**e**) ε = 0.125, and (**f**) ε = 0.125. The atoms are colored according to the CNA method and Dislocation Analysis (DXA) algorithm.

**Figure 6 materials-13-02803-f006:**
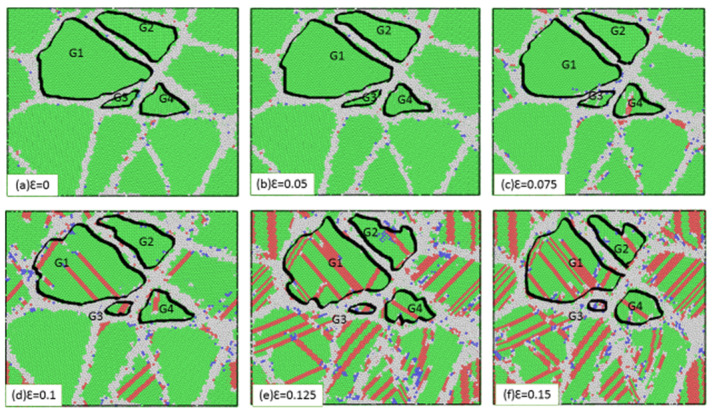
Snapshots of stainless steel polycrystalline samples with an average grain size of 3.6 nm at a temperature of 300 K: (**a**) ε = 0.0, (**b**) ε = 0.05, (**c**) ε = 0.075, (**d**) ε = 0.1, (**e**) ε = 0.125, and (**f**) ε = 0.15.

**Figure 7 materials-13-02803-f007:**
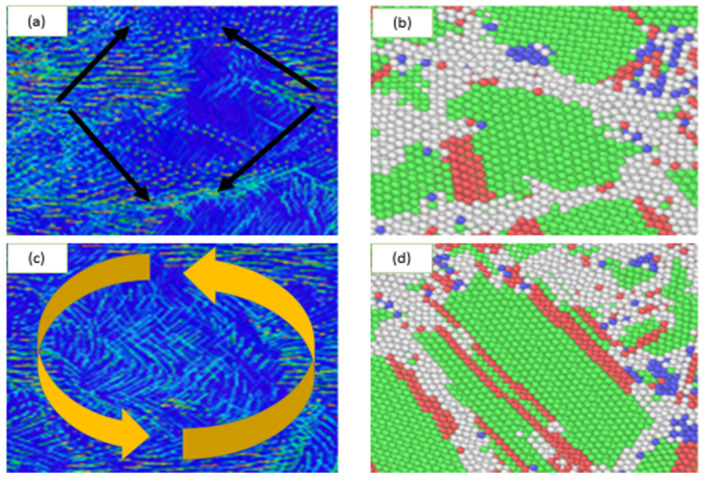
Snapshots showing (**a**,**b**) grain boundary sliding and (**c**,**d**) grain rotation for the sample with an average grain size of 3.6 nm. (**a**,**c**) were generated using atomic displacement vectors to visualize the grain boundary sliding and grain rotation, respectively. (**b**,**d**) were generated using the CNA method to visualize the grain boundary regions.

**Figure 8 materials-13-02803-f008:**
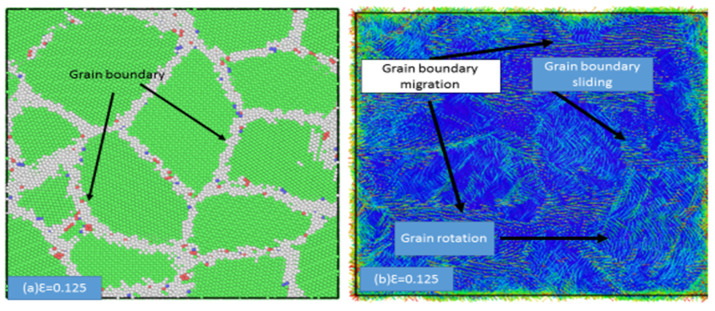
Snapshots of the atomic displacement vectors for the sample with an average grain size of 2.5 nm, depicting (**a**) the grain boundary at ε = 0.02 and (**b**) grain sliding, grain rotation, and grain migration at ε = 0.125.

**Figure 9 materials-13-02803-f009:**
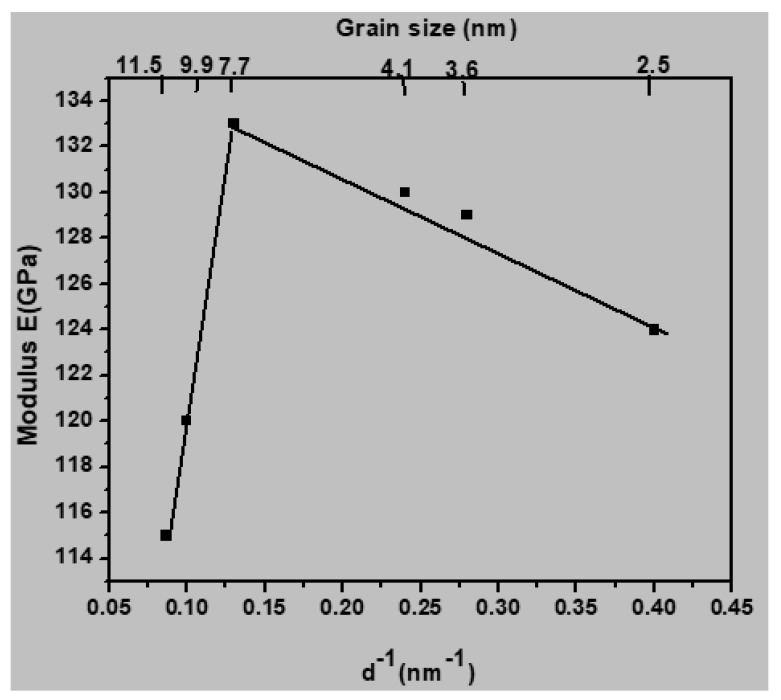
Young’s modulus *E* as a function of reciprocal grain size *d*^−1^ at 300 K.

**Figure 10 materials-13-02803-f010:**
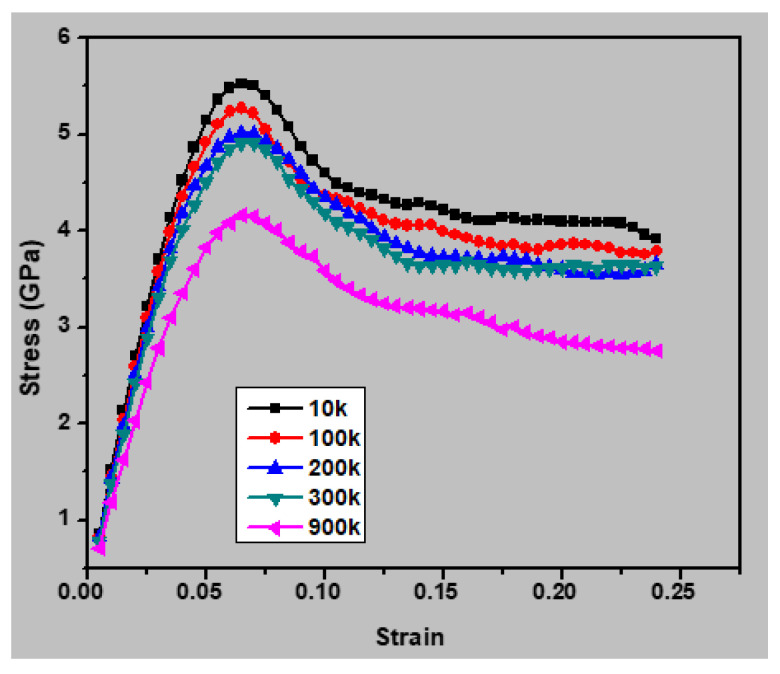
Stress–true strain curves for the nanocrystalline austenitic stainless steel sample with an average grain size of 2.5 nm at various temperatures.

**Figure 11 materials-13-02803-f011:**
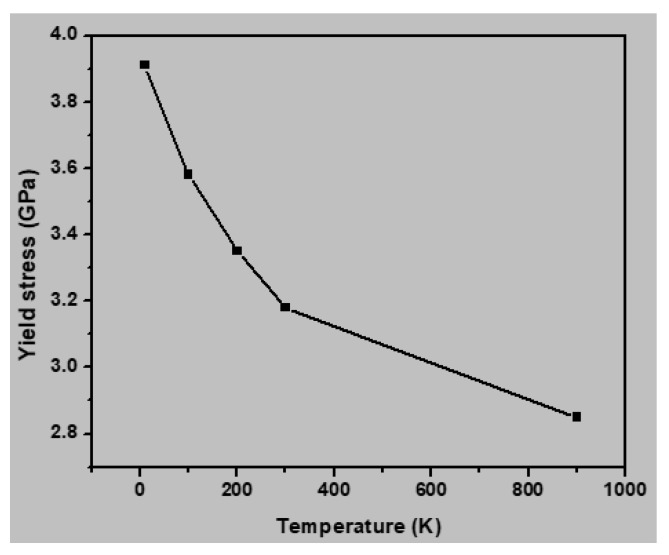
Yield stress of the nanocrystalline austenitic stainless steel sample with an average grain size of 2.5 nm as a function of temperature.

**Figure 12 materials-13-02803-f012:**
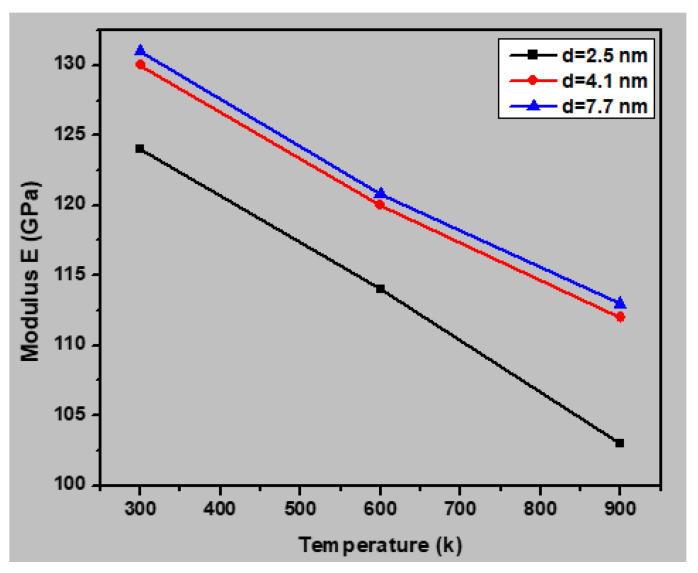
Young’s modulus of the nanocrystalline stainless steel samples with average grain sizes of 2.5, 4.1, and 7.7 nm with respect to temperature.

**Figure 13 materials-13-02803-f013:**
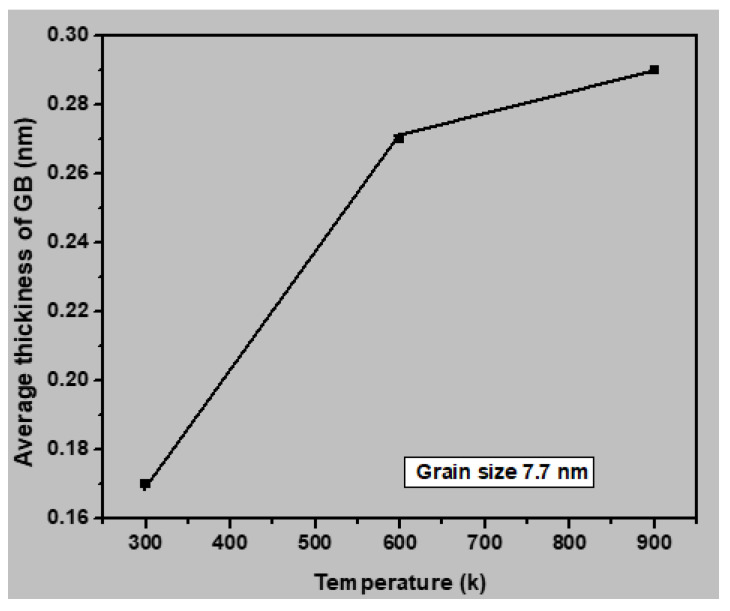
Average grain boundary thickness for the nanocrystalline stainless steel sample with an average grain size of 7.7 nm with respect to temperature (300, 600, or 900 K).

**Figure 14 materials-13-02803-f014:**
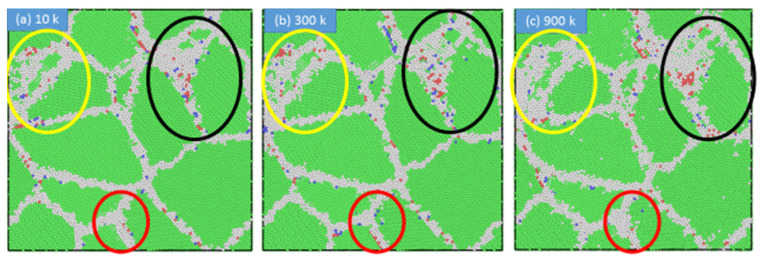
Snapshots showing the influence of temperature on the grain boundary thickness for the stainless steel sample with an average grain size of 3.6 nm: (**a**) 10 K, (**b**) 300 K, and (**c**) 900 K.

**Figure 15 materials-13-02803-f015:**
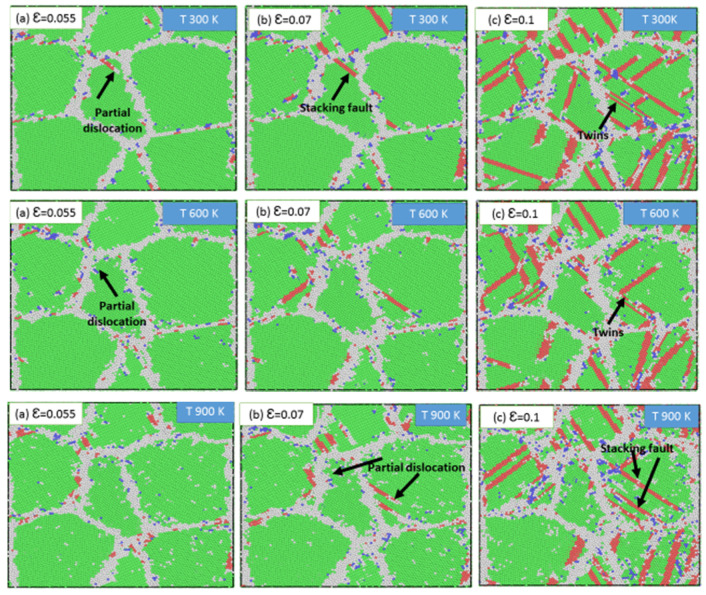
Snapshots of the sample with an average grain size of 7.7 nm at various temperatures 300, 600, and 900 K, and various strain (**a**) *ε* = 0.055, (**b**) *ε* = 0.07, and (**c**) *ε* = 0.1.

**Figure 16 materials-13-02803-f016:**
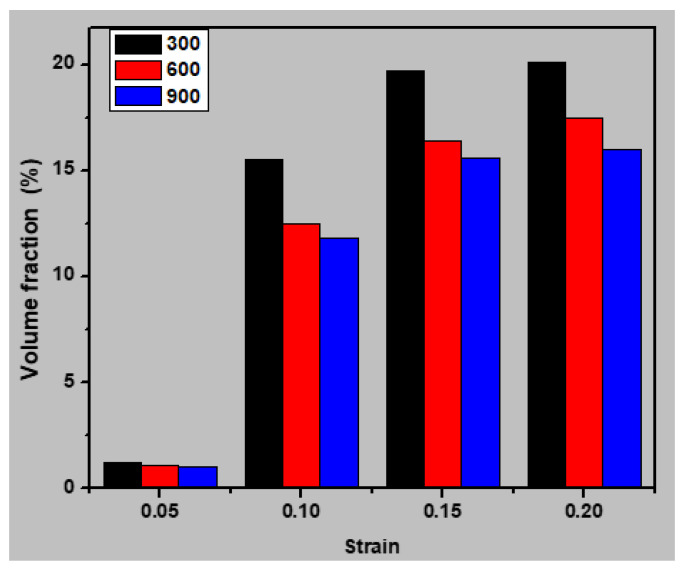
Volume fraction of twins and stacking faults (hcp) in the sample with an average grain size of 7.7 nm for various strains and temperatures (300, 600, and 900 K).

**Figure 17 materials-13-02803-f017:**
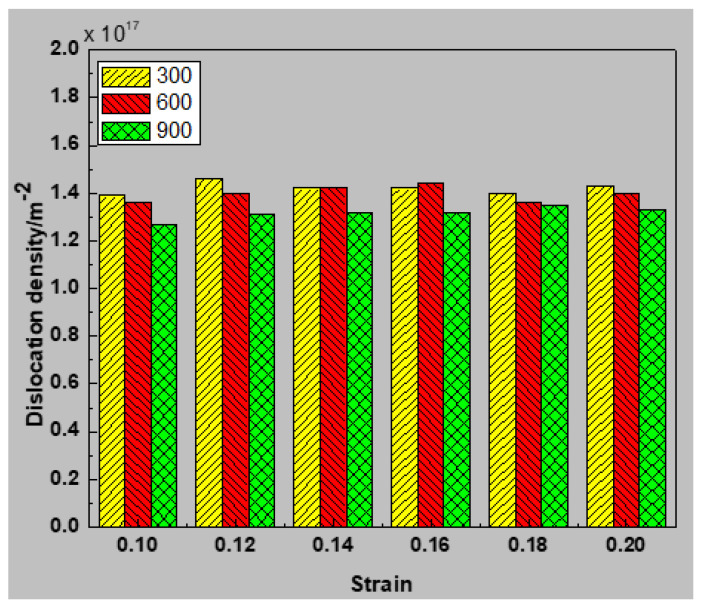
Variation of dislocation density of sample with a mean grain size of 7.7 nm at temperatures (300, 600, and 600 K).

**Table 1 materials-13-02803-t001:** Mechanical properties of the nanocrystalline austenitic stainless steel samples at room temperature.

Grain Size (nm)	Tensile Strength (GPa)	Yield Strength (GPa)	Young’s Modulus (GPa)
11.5	5.44	3.7	120
9.9	5.6	3.8	115
7.7	5.65	4.2	131
4.1	5.1	3.9	130
3.6	4.8	3.75	129
2.5	4.9	3.5	124

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
