# Peer review of "Influence of Temperature on Mechanical Properties of Nanocrystalline 316L Stainless Steel Investigated via Molecular Dynamics Simulations"

_materials, 2020, doi:10.3390/ma13122803_

Round 1

Reviewer 1 Report

The authors use molecular dynamic modelling to study the mechanical properties of nanocrystalline 316L stainless steel. The stress-strain curves of the investigated material for grain sizes ranging from 2.5 to 11.5 nm were  constructed including the temperature effect (10-900K). It was found that the Hall-Petch type of relation is valid for grains larger than 7.7nm, for smaller grains an inverse relation was found. Also Elastic modulus showed different behavior above and below 7.7nm of grain size. Various types of crystal defects were identified and their role in the deformation mechanism was described. It was observed, that the grain boundary sliding and grain rotation are responsible for plasticity of materials with <7.7nm grain size, whereas the larger grain size specimens deformed by twinning and dislocation processes.

The article has easy to follow structure and is written in good English containing relevant references. The topic is of a high relevance, and the results are interesting. However the article should be revised to make the presentation of results more clear so that the conclusions follow from the results more explicitly. Therefore MINOR REVISION is recommended by the reviewer.

Major comments:

  1. Most of the findings follow from the graphic representation of the atomic structure in forms of common neighbor analysis (CAN) and (DXA-dislocation extraction algorithm , definition missing?) . These charts should be presented in unified way (i.e. Figure 4 should match figs 5 6 and 7, grain boundary image on figure 8a should be explained etc.). Both CAN and DXA results should be presented for different grain sizes (figs 5 and 6). Then is can be easily seen whether there is less dislocations in small grain sizes.
  2. Similarly to the point above, grain boundary sliding and grain rotation should be presented on both smaller and larger grain sizes (so far only 2.5nm and 3.6nm is presented) to see the difference. The displacement map is not very clear at the moment and should be improved.
  3. The grain size seemingly does not match the atomic scale (raster) in the figures, putting a scale bar might help. It is also hard to see 5nm mean grain size in Fig 1 (4-6 grains per 200nm ~5nmgrain size), what is the grain size distribution ?
  4. The obtained mechanical properties should be discussed in more detain for example the seemingly low elastic modulus may be the result of nonlinear elasticity. At the same time, the procedure of extraction of E from loading curve might play a role and should be described.

Minor comments:

Fig. 2 – Gpa->GPa

L152 – What mechanism would hinder the partial dislocations, the presence of partial dislocations at the boundary ? Not clear from text.

L174 – emit ? enter ? travel ?

L205 – “Furthermore, the rotation and migration of the grain boundaries and the rotation inside of the atoms.“-> can be observed ? Include and refernce displacement map ? Rotation inside the atoms ?????

Fig. 8a – what kinf of contrast is that ?

L238 – “To get a fully dense material in the flexible system, the relationship between the Young’s modulus and the applied stress and strain is defined by Hooke’s law.“ ????? To get fully dense material ???

L251 – „However, the relationship between E and grain size tends to be more suitable for nanocrystalline Fe-Cr-Ni austenitic stainless steel in the present study, at least in the range of average grain sizes between 3.6 and 7.7 nm.“  – rephrase such as „This investigated Fe-Cr-Ni steel follwed the above relationship in range between 3.6-7.7mm.

L266 – „…..temperatures. Consequently, a higher temperature increased the deformation resistance of the materials, thereby decreasing the Young’s modulus. ????“  Elastic modulus decreases with temperature as expected . What si deformation resistance ? Stifness ?

Fig. 16 – x-axis is misleadin, Supposedly same strain ws applied for all three teperatures !? Note that similar typ charts may quantitativelly support the findings of the paper, i.e. the mechanism of plasticity difference below and above 7.7nm, I recommend to include these charts (dislocation density, grain boundary slip and rotations ) it in the paper.

L328 – Hall-Petch DOWN TO critical grain size of 7.7nm

Author Response

Response to Reviewer 1 Comments

Reviewer 2 Report

In this manuscript, general mechanical properties of nanocrystalline stainless alloy were modeled by molecular dynamics simulations. The whole study is based on simulation and modelling results without any experimental verification. However, the manuscript is well written, and the results are satisfyingly presented.

Still, some explanations need to be provided before the acceptance:

  • - From metallurgical point of view, a Fe alloy without carbon element is not a steel. In principle, the properties of metallic alloy depend on microstructure and microstructure depends on chemical composition. How is the resulted microstructure (or nanostructure in this case) affected, if the carbon is not incorporated into the modelling? Are the resulted properties accurate, if such important element (for steels) was omitted? Authors have to provide at least some explanation, why their results are not deteriorated by this issue.
  • - I would recommend changing Angstrom to nm.
  • - L 101-105: The system was relaxed under a pressure of 0 bar, but what was the pressure during the simulated tensile deformation?
  • - Tab. 1: Why is the elastic modulus so low at grain size 9.9 nm?
  • - Correct units for stress (GPa instead of Gpa) in all graphs.
  • - Fig. 5: What is the difference between Figures (a) and (b), between (c) and (d) and between (e) and (f)? Because the strains are the same for each group. In the related text is as the first mentioned Figure 5(c). The explanation should start with Figure 5(a).

Author Response

Response to Reviewer 2 Comments

Reviewer 3 Report

The topic of the manuscript entitled “Influence of temperature on mechanical properties of nanocrystalline 316L stainless steel investigated via molecular dynamics simulations” falls within the scope of Materials. The paper contains interesting experimental results and corresponding analyses. It is of sufficient scientific interest and has originality in its technical content to merit publication. The authors have cited the relevant literature. Methods and interpretations of results are correct and novel. The issues were well presented. In terms of content, the analysis does not raise any objections. The arrangement of work maintains substantive continuity and constitutes a logical whole.

However, the manuscript is not suitable for publication in its present form. This paper requires minor corrections.

Comments and remarks are presented below.

Reference numbers in the text should be immediately after the names of the publication authors, and not at the end of the sentence.

Hooke’s law (page 9) should be on one line and should be numbered (although this is the only equation in the manuscript).

The legend of the bottom axis in Figure 9 is incorrect.

Author Response

Response to Reviewer 3 Comments

Round 2

Reviewer 2 Report

All questions were clarified by the authors. The manuscript was improved. Therefore, I recommend publication of the presented manuscript.